# Nutrition for European Elite Fencers: A Practical Tool for Coaches and Athletes

**DOI:** 10.3390/nu16081104

**Published:** 2024-04-09

**Authors:** Marta Lomazzi

**Affiliations:** Institute of Global Health, University of Geneva, 1202 Geneva, Switzerland; marta.lomazzi@unige.ch

**Keywords:** fencing, sports’ nutrition, performance, injury recovery, supplements

## Abstract

The aim of this narrative review is to create a comprehensive, innovative, and pragmatic resource to guide elite fencers and coaches in making strategic nutritional choices to enhance performance and facilitate recovery. The literature review identified only 12 articles specifically addressing the topic of nutrition for fencers. Thus, the recommendations provided in this review derive also from articles dealing with similar sports, such as martial arts, and from investigations with European elite fencers and their coaches. For elite fencers, it is suggested to consume daily 7–11 g/kg of body weight (BW) of carbohydrates and 1.5–2 g/kg of BW of proteins and allocate 25% to 30% of the total energy intake to essential fats, with a specific focus on omega-3 fatty acids. The timing of meals, ideally within one hour after exertion, plays a pivotal role in restoring glycogen reserves and preventing injuries. The intake of leucine, creatine, omega-3, collagen, and vitamins C and D is proposed as a strategy for injury recovery. It is worth acknowledging that even when personalized plans are provided, implementation can be challenging, especially during competitions and training camps.

## 1. The Art of Fencing

The sport of fencing is both physically and mentally demanding, encompassing offensive and defensive actions using a sword [1]. Functioning as a martial sport, it necessitates swift reflexes, agility, explosivity, endurance, and mental sharpness. Fencing typically involves a sequence of powerful attacks intertwined with moments of low-intensity movements and recovery phases, primarily sustained by anaerobic metabolism. The metabolic demand is complex and multifaceted, reflecting the sport’s intense, burst-like nature combined with periods of strategic maneuvering. The activity would predominantly rely on the anaerobic system during bouts of rapid, explosive movements, such as lunges and parries, requiring high levels of muscular strength, power, and speed. Simultaneously, aerobic capacity cannot be overlooked, as it supports sustained activity and recovery between bouts. The sport’s unique blend of dynamic physical engagement, mental strategy, and technical skill underscores the importance of a comprehensive training program designed to meet these diverse metabolic requirements [2,3]. This training journey should be complemented and upheld by appropriate nutritional practices to ensure adequate energy intake, replace fluid losses due to the high sweat rate, and facilitate recovery. Throughout its evolution, fencing has captured interest from coaches and competitors, resulting in an expanding body of literature concentrating on training approaches, but only partially about optimizing dietary tactics to amplify performance and facilitate recovery. The aim of this review is to provide a novel and comprehensive nutritional tool to assist coaches and elite athletes in their endeavors. The recommendations presented are grounded in the scientific literature and real-world situations, drawing from conversations with elite athletes and coaches, mainly from Italian and Estonian Olympic teams, with whom the author has extensively collaborated over the years. These insights aid in identifying effective nutritional approaches to support the athlete in reaching his/her optimal performance across diverse contexts, namely during training, competitions, and recovery.

## 2. Materials and Methods

This narrative review encompasses an examination of various studies in the literature, focusing on the field of sport nutrition for elite fencers. The literature review was performed on the PubMed engine, using the following keywords (and related Medical Subject Headings—MeSH—when possible): (((fencing[Title/Abstract]) OR (fencers[Title/Abstract])) AND (nutrition OR diet OR supplements))). Before arriving at this final string, several keywords’ compositions were tested to define the most relevant and complete string in relation to the research question. Articles published in English up to July 2023 and containing information on nutrition for elite fencers in the title or abstract were included. The initial list of 52 articles was further checked for redundancy (duplicate articles), invalidity (the incorrect target can primarily be attributed to the dual interpretation of “fencing”, encompassing both the sport and agricultural connotations), and topic relevance. Many articles focused on the training aspects of fencing, while very few provided recommendations on nutrition strategies for elite fencers. Only 12 manuscripts were selected for further investigation, which underlines the need for more research in this area. The PubMed search was also combined with a Google Scholar search using the same keywords. In addition, statements from recognized associations such as IOC Consensus Statements were reviewed and included in the analyses where relevant. When appropriate, the recommendations presented in this review are adapted from articles centered on sports that share analogous characteristics, such as martial arts. This review also incorporates the researcher’s personal experiences and inputs from sport coaches and elite fencers, with whom the author has worked for several years. These contributions are explicitly indicated within the text using quotation marks, accompanied by proper source citations. 

## 3. Training and Competition Schedules

The competition schedule for elite fencers in Europe can vary based on factors such as the specific fencing discipline (foil, epee, and saber), the fencer’s national and international ranking, and the time of year. However, there are some common elements in an elite fencer’s competition schedule. In general, the season starts around mid-November with the World Cup stages and concludes in June/July with the European and World Championships or potentially the Olympics. During the season, athletes also participate in various national competitions. The season is long and demanding, with the first peak occurring between late January and mid-April (World Cup stages) and the second peak in July (World Championship or Olympics) [4,5]. Athletes typically reach their optimal physical condition around January.

Athletes usually train twice per day, on five or six days throughout the week, with different coaches, such as a personal trainer, respiratory trainer, and fencing master. The annual training program varies based on the athletes’ needs and the schedule of their competitions (not all athletes participate in all events). From September to November, the focus is on the loading phase, characterized by higher training volume and density. Athletes focus on aerobic endurance, lactic endurance, and strength through workouts that develop overall athletic abilities. Then, leading up to the end of the World Cup, the emphasis shifts to specialization, involving high-intensity but lower-volume training to facilitate recovery and maintain a high level of specific intensity, closely resembling what happens during competitions. Depending on summer commitments, a brief loading phase is reprogrammed, followed by another phase of specific intensity training. There are two main recovery phases scheduled towards the end of December/beginning of January, and in April.

A common weekly training schedule is organized around three lessons of 30 to 40 min each with the coach, three sessions of free or themed bouts, and three sessions of training for respiratory and ocular muscles, maintaining high attention and concentration levels, specific endurance, and precision, plus two or three sessions of traditional workouts (schedule drawn from the usual weekly training schedules of several European fencing teams). The training is periodized and tailored to the individual. Workload management is a critical aspect of fencing training, as excessive training loads can lead to fatigue, injuries, and performance stagnation. According to Andrea Vivian, respiratory trainer of several European fencing teams’ elite athletes, “recovery cycles are essential in fencing to obtain a psychophysical recharge and to have fresh ideas to take to the strip”. 

Fencing tournaments encompass a full day, commonly extending for approximately 10 h, and include roughly 10 bouts, with intervals ranging from 15 to 300 min between each bout. The bouts and the actual fighting time constitute just 13% and 5% of the total competition duration, respectively. Typically, a foil fencer initiates attacks lasting about 5 s, predominantly relying on the phosphagen system to meet the immediate ATP requirements, while an épée fencer does so for approximately 15 s, drawing significantly on anaerobic glycolysis due to their longer duration, with a significant portion being submaximal, prior to each period of rest or break. Additionally, during a bout, fencers may cover distances ranging from 250 to 1000 m [3].

## 4. Stress and Fatigue

Much like other combat sports, engaging in fencing trainings and competitions results in fatigue [6,7]. Fatigue is a sensation conventionally linked with heightened perceptions of weariness and diminished energy levels, stemming from either physical, mental, or a combination of physical and mental exertion. During competitions, the successive bouts carried out within a single day pose a considerable psychophysical challenge for athletes, particularly during the afternoon or evening sessions. This period witnesses a decline in both physical and mental capacities owing to overall fatigue and extended stress, in addition to the important effort required to win a bout [8,9]. A fencing competition can induce an increased perception of fatigue associated with an important perceived mental demand, which intensifies throughout the competition; this implies that elite fencers must channel their mental resources more extensively than their physical ones to counteract the negative influence of exhaustion on performance [9]. As discussed in the next paragraph, nutrition assumes a pivotal role in ensuring sufficient energy for the entirety of the competition. Nonetheless, care must be taken to ensure that a tailored nutritional approach for athletes does not exacerbate the burden on an individual already carrying a substantial mental load. According to Andrea Vivian, “all athletes who have relied on a nutritionist had important improvements, both in terms of anthropometric measures and performance. However, in a few cases I have noticed numerous psychological challenges in adhering to the recommendations provided by the nutritionist, leading to concentration difficulties and a decline in performance”. On the other hand, Dario Chiado, Technical Director of the Italian National Fencing Team, underlines that “an athlete who pays attention to every detail, including nutrition, is psychologically stronger and finds the correct interpretation of the work, without impacting overall psychological stress”. When discussing with elite athletes such as Andrea Santarelli, winner of a silver medal at the Olympic games in Rio de Janeiro 2016 and of a gold medal at the World Championship in Milano 2023, the importance of nutrition appears evident: “nutrition is a very important part of an athlete’s career. With proper nutritional advice, I’ve understood very well what makes my body function harmoniously and what I could define as ‘harmful’. From this moment on, since fencing is a highly psychological sport, I strive to maintain the right balance between personal satisfaction and physical fitness in order to face the competition moments at my best”. This feeling is also shared by Enrico Garozzo, winner of a silver medal at the Olympic games in Rio de Janeiro 2016 and of a bronze medal at the World Championship in Khazan 2014: “at a psychological level, following a diet has formed new habits that I really enjoy. Stress used to occur only at the beginning of the diet, now stress arises if I do not follow it. It is a new *forma mentis*”.

## 5. Nutrition for Elite Fencers

Nutrition holds a central role in fencing performance, impacting energy levels, recovery, cognitive function, and muscle growth. However, fencing athletes often lack sufficient knowledge about appropriate dietary practices. Numerous dietary patterns involve a substantial intake of processed and refined foods, accompanied by a notable consumption of saturated fats, while fresh fruits and vegetables are consumed in insufficient quantities. These dietary habits negatively affect overall health and present challenges to athletic performance [10]. In general, the Italian fencing societies do not offer individualized nutritional guidance; instead, they provide general recommendations during annual seminars, and the coaches try to provide suggestions. “There are occasions where I find it challenging to convince athletes to consume food during competitions; the idea of participating in a day of competition without eating appears unreasonable, yet it occurs frequently when athletes lack guidance from a nutritionist”, reveals Andrea Candiani, former Technical Director of the Italian National Fencing Team. He further explains, “when fencers are in their early stages, it is crucial to help them grasp the importance of proper nutrition”. Top-level athletes, realizing the significant impact of nutrition on performance and recovery, typically seek out a sport nutritionist or dietitian for personalized counseling. According to Dario Chiado, “physical performance is progressively emerging as a decisive factor. Nutrition is as essential as physiotherapy; it must be integrated into the planning. Some athletes have an absolute requirement for it, while others need it less. It is not something to be overlooked for anyone”. This perception is also supported by Stefano Campari, Fitness Coach of several elite fencers: “I believe that nutrition is a fundamental piece in an athlete’s preparation. I think it should be as personalized as possible, based on the athlete’s physiology and performance characteristics”. To note, he also underlines, “I have seen significant improvements in the performance of most athletes in terms of energy levels and better recovery quality. However, for a small percentage, I have noticed a decline. Certainly, this small percentage struggled to cope with the stress of the diet. Yet, a deeper analysis leads me to state that athletes who naturally possess a favorable weight-to-power ratio, which results in explosive and rapid fencing characterized by powerful movements, tend to experience a decline in performance when brought to reduce weight”. Within this context, the role of the sport nutritionist is crucial in shaping the personalized approach that leads to enhanced performance when and as needed.

### 5.1. Body Composition

The significance of body weight and body composition is closely tied to an athlete’s performance [10,11,12]. While the ideal body composition can vary based on the sport, in general, a lower fat mass is correlated with greater performance potential. Although certain studies have suggested that fencing success is more dependent on factors such as skills, rapidity, and agility rather than a high aerobic capacity and low body fat percentage, there is evidence demonstrating that aerobic training can ameliorate fencers’ reaction times, increase their attention span, and lead to a decrease in body fat [13,14]. Within this context, the sport nutritionist should first assess the body mass of the elite athlete and support him/her in reaching the ideal body weight and composition. As highlighted by Enrico Garozzo, “on the fencing strip, having lost a significant amount of weight, I was lighter, resulting in fewer impacts on the knees and ankles, and an increased performance”. 

### 5.2. Nutrients

As highlighted above, elite fencers’ training is structured around periods, where the composition of exercise is strategically modified within a series of cycles. Similarly, dietary intake and nutrition strategies should be consistently adapted to the continuously evolving training program, aligning with the micro-, meso-, and macro-cycle training, all aimed at ultimately achieving long-term objectives [15]. Strategies for nutrition during competitions could span pre-event, in-event, and inter-event eating, all carefully devised to meet the requirements for fuel and fluid replenishment [16]. While fencing is primarily an anaerobic activity, maintaining aerobic fitness is essential to undergo the numerous bouts needed to advance to the tournament finals. The extent of oxidative involvement ranges between 81% (during the initial bout) and 90% (within the subsequent bout). This is trailed by a contribution of 9% (in the second bout) to 12% (in the first bout) from the ATP-PCr system, and a contribution ranging from 0.6% to 7% from the glycolytic system in the context of three sets of 3-minute épée match simulations [17,18]. The energy demand is influenced by factors such as gender, age, training ability, and the specific technical and tactical strategies employed in response to the opponent [3]. During a tournament, fencers cover long distances characterized by a combination of high-intensity actions alongside submaximal actions, which have the potential to induce neuromuscular fatigue. Elevated levels of creatine kinase (CK) immediately after, as well as 24 and 48 h following a tournament, combined with a rise in lactate dehydrogenase (LDH) levels right after the event, suggest that the physical demands of the tournament can cause muscle stress and damage [19]. The average oxygen consumption (VO2) during competitions reaches 53.9 mL/kg/min for male fencers and 39.6 mL/kg/min for female fencers. These VO2 demands necessitate a comprehensive conditioning of both anaerobic and aerobic metabolism for competitive fencers [20]. Therefore, dietary strategies must be carefully crafted to meet the specific demands. In the context of elite fencers and beyond, fundamental principles dictate that the composition of macronutrients, meal timing, and maintaining proper hydration all contribute significantly to improving fencing performance and supporting recovery. It is essential to strategically include carbohydrates in the diet, given their role as the main energy source for intense activities. This helps in replenishing glycogen reserves and maintaining energy levels throughout both training sessions and competitive events. Furthermore, adequate protein consumption is indispensable for repairing and promoting muscle growth, particularly due to the repetitive and demanding motions inherent in the sport of fencing [21]. 

In sports characterized by high intensity, the recommended carbohydrate intake can be around 10–12 g per kilogram of body mass daily [22]. Fencers are advised to consume about 7 to 11 g per kilogram of body mass daily, focusing on complex sources like whole grains, fruits, and vegetables [23,24]. During the loading phase from September to November, which involves increased training volume, carbohydrate consumption might need to be slightly elevated to meet the heightened demand. Opting for low-glycemic-index carbohydrates has been proposed to enhance training endurance compared to high-glycemic-index products [25]. However, athletes engaging in multiple high-intensity training sessions per day should opt for carbs with a high glycemic index to speed up muscle glycogen replenishment before the second session [24]. Pre-exercise, the consumption of carbohydrates in the hours leading up to exercise can not only improve performance in endurance sports but also in strength and power activities that entail repeated high-intensity efforts as fencing does. This strategy could potentially serve as a means of supporting fencers during competitions [26]; the practice of briefly rinsing the mouth with a sports drink for approximately 10 s before a competition could be an interesting approach, especially when athletes might opt against consuming foods or fluids [23]. Post-exercise carbohydrate consumption is essential for enhancing recovery and restoring glycogen levels depleted during training, and high-glycemic products might facilitate faster restoration of muscle and liver glycogen stores compared to complex carbohydrates [27]. General guidelines for glycogen restoration span from 5 to 7 g/kg/day for athletes involved in moderate-volume training, while a higher intake of 7 to 10 g/kg/day is advised to thoroughly replenish glycogen stores [28]. For optimal muscle glycogen replenishment in the short-term recovery phase, it is crucial to promptly ingest a carbohydrate supplement right after exercise. When relying solely on carbohydrate intake, supplementation should be carried out at regular intervals, like every 30 min, providing around 1.2 to 1.5 g of carbohydrates per kilogram of body weight per hour. Nevertheless, incorporating protein into a carbohydrate supplement can substantially improve the effectiveness of muscle glycogen storage. This approach reduces the necessary amount of carbohydrate intake and the frequency of supplementation required for achieving optimal glycogen storage [22,29]. If both carbohydrates and protein are consumed, it is recommended to have an immediate intake of 0.8 g of carbohydrates per kilogram of body weight, along with 0.2 g of protein per kilogram of body weight, followed by another ingestion two hours post-exercise within a four-hour recovery timeframe [29]. On top of the recommendations provided, understanding the glycemic impact of consuming large quantities of carbohydrates, even if they are wholemeal, is crucial for athletes’ optimal performance and health. Wholemeal carbohydrates, while generally healthier due to their higher fiber content and slower digestion rate, can still lead to significant increases in blood sugar levels when consumed in large amounts. This can cause fluctuations in energy levels, potentially leading to periods of high energy followed by sudden drops, which might affect training effectiveness, recovery, and overall performance. Therefore, it is important for athletes to balance carbohydrate intake, considering both the quantity and the type, to maintain stable glycemic levels, ensuring sustained energy and peak performance. In my work with elite fencing athletes over the years, I have consistently emphasized the preference for whole grains, particularly rye, together with fresh and dried fruit, as source of carbs, to ensure a sustained supply of energy and prevent drowsiness after meals. 

Regarding protein, fencers should aim for approximately 1.5 to 2 g per kilogram of body weight daily, favoring lean sources like chicken, fish, eggs, dairy products, tofu, and legumes. Athletes aiming to lose weight may require a higher protein intake, typically recommended at 1.8 to 2.7 g per kilogram of body mass, to mitigate the risk of muscle loss during periods of energy deficit [23,24,30]. Meats, particularly red meats, have long faced criticism for their content of saturated fat. However, meats also encompass significant quantities of monounsaturated fatty acids such as conjugated linoleic acid which are important for athletes and can lead to a reduction of fatigue and increase exercise outcome [31,32]. 

The suggested fat intake for athletes’ diet ranges from 25% to 30% of their energy intake, with the preferred sources being unsaturated fatty acids, mostly omega-3 fatty acids, which reduce inflammation and oxidative stress, ease muscle soreness, and improve muscle protein synthesis [33]. Thus, incorporating healthy fats such as nuts; seeds; vegetable oils, such as olive oil; sea fish; and avocados is essential for energy storage, as well as for feeling satiated.

Beyond macronutrients, it is crucial for fencers to consider their intake of micronutrients. Adequate consumption of vitamins and minerals is necessary to bolster the immune system, maintain bone health, and promote overall well-being. Foods such as leafy greens, nuts, seeds, and berries are rich in antioxidants and vitamins that aid fencers in recovering from rigorous training sessions and competitions [1,34]. 

Regardless of the nutritional approach adopted, it is recommended that the potential for gastrointestinal discomfort is considered. Opting for carbohydrates that are easily digestible, especially those with low fat content, minimal fiber, and reduced gas-producing potential, appears to be a sensible approach, especially during competitions.

### 5.3. Meals’ Timing

Meal timing plays a crucial role. Fencing athletes typically adhere to a daily meal schedule consisting of three or four meals. As previously emphasized, while various strategies exist for replenishing glycogen stores after exertion, it is paramount to promptly ingest carbohydrates and protein together after exercise. In my professional experience, I try to translate the scientific evidence into recommendations that are easy to handle and put into practice for elite fencers, who very often undertake university studies in addition to their fencing activities. I advise athletes to consume their meal within 60 min after workout, maintaining a ratio of roughly 4 parts carbohydrate to 1 part protein. Among the elite fencers I collaborate with, meals generally adhere to these guidelines, with lunch and dinner being very close to the end of the morning or afternoon training sessions. In situations where this is not feasible, it is recommended to consume a snack composed of carbohydrates and proteins post-training, such as whole grain bread and cheese or milk or dried meat with an apple. During competitions, athletes often lack the opportunity for proper meals and instead opt for quick, small snacks. In my experience, incorporating mini meals, several times during the competition day, containing rye bread, lean meats or cheese, fruits, and dried fruits serves as a practical and effective approach to sustaining high energy levels while warding off fatigue during bouts. Practical nutritional guidelines for elite fencers outlined in this article are summarized in Table 1.

### 5.4. Hydration

Hydration is an essential ergogenic tool for athletes, as maintaining proper hydration levels is vital for sustaining exercise capacity both during training and on the day of competition. Athletes are advised to consume approximately 0.5 to 2 L of fluids per hour during physical activity, with a baseline intake of at least 35 mL per kg of body weight daily when at rest. Moreover, athletes can consider drinking 200–300 mL of fluids 20–30 min before any competition, tailoring this to their unique sweat rates and the prevailing environmental conditions, aiming to maintain steady hydration, while avoiding gastrointestinal discomfort. The choice between water and glucose–electrolyte solutions should be made based on individual hydration and energy needs [35] (Table 1).

Dehydration is a major enemy of fencers’ performance and health. Dehydration levels of 2% can negatively impact athletic performance, while dehydration reaching 4% can result in severe health consequence [24]. Achieving ideal hydration depends on numerous variables, such as sweat rate, environmental condition, type and intensity of the activity, etc., but it is generally characterized by preventing losses that exceed 2–3% of body mass during exercise [1,36]. In fencing, the use of protective gear limits evaporative heat loss and intensifies the thermal regulation requirements, leading to increased demands for hydration. Nonetheless, sweat rates are typically modest during competitive fencing. In regular tournament settings, fencers have ample chances to adequately rehydrate and counterbalance fluid depletion [3,6,20]. Athletes and coaches may consider monitoring body hydration levels before and after competitions. Utilizing tools like the Beverage Hydration Index (BHI) can provide precise measurements, ensuring that athletes maintain optimal hydration to support their physical exertion and strategic execution in fencing [37].

### 5.5. Supplements

Various renowned international institutions periodically issue official statements to assess the level of evidence for sports supplements [38]. According to Mata and colleagues, medical professionals exhibited a higher tendency to recommend supplement consumption to men fencers, while friends were the primary source of such advice for women. The prevalence of sport supplement usage within the fencing community mirrors earlier findings among athletes with comparable competitive standings [34,39]. Elite fencers predominantly utilize sports drinks, bars, caffeine, vitamin C, and beetroot juice as their primary supplements, mainly to boost cognitive performance [40,41,42]. Caffeine is among the most popular supplements in sports, readily available in various convenient forms, including chewing gum, capsules, and energy drinks. In elite sports, it is used for its performance-enhancing effects, which are attributed to a range of physiological and psychological mechanisms. Caffeine intake may range from 3 to 6 mg/kg body weight to be taken about 60 min before the competition/exercise [43]. Concerning vitamin C, the optimal dosage depends on the intensity and duration of the workouts. Generally, a daily intake of 500 to 2000 mg of vitamin C is advised to support overall health and minimize oxidative stress [44]. It is generally advised to consistently consume vitamin C for several weeks leading up to a competition [44]. Beetroot juice, due to its high content of dietary nitrates, can enhance athletic performance. Supplementation of around 5–9 mmol of nitrate per day for a period of 1–15 days can lead to positive impacts on physiological exercise responses [45]. In addition, athletes may consume whey protein and iron, with the latter being very important especially for female athletes [46,47]. Other supplements that athletes might consider include beta-carotene and creatine. For creatine, the standard recommendation for athletes looking to improve performance and aid in muscle recovery is to follow a loading phase of 20 g per day for 5–7 days, followed by a maintenance phase of 3–5 g per day. Beta-carotene and creatine have been suggested to potentially enhance cognitive performance, although more research is necessary to definitively establish their effectiveness in supporting cognitive function [48,49]. Lastly, vitamin B12 plays a crucial role in producing red blood cells for oxygen transport, enhancing energy metabolism, and facilitating muscle recovery. The recommended dietary allowance (RDA) for vitamin B12 is about 2.4 micrograms per day. However, needs can vary based on an individual’s dietary pattern, absorption rates, and overall health. Predominantly sourced from animal products, B12 supplementation becomes particularly important for vegetarian or vegan athletes to prevent deficiency and maintain peak performance [50].

Before delving into discussions about supplements with athletes, it is crucial to ensure that they adhere to a well-balanced diet. Often, a proper diet can provide adequate amounts of the nutrients that some supplements offer, making additional supplementation unnecessary or potentially harmful. It remains crucial for both athletes and their nutrition experts to conduct a cost–benefit evaluation regarding their judicious and responsible utilization [38,51]. 

## 6. Practical Constrains for Optimal Nutrition Implementation

As highlighted by most of the elite fencing athletes the author has worked with over the years, the main practical constraints to implementing nutrition strategies are linked to travel within and outside Europe to attend the competitions. The same applies to training camps, even if within their home country. “It is very difficult to follow proper nutritional advice during competition or camps because there is no kitchen or if there is, it is limited; the restaurant isn’t good, so we have to go to the supermarket to buy ham, dried meat, etc. which isn’t optimal on the long term”, says Erika Kirpu, winner of a gold medal at the Olympics in Tokyo 2020 and at the World Championship in Lipsia 2017. “I brought a steamer to Tokyo Olympics to prepare healthy foods. The difficulty is organizing it the first time, then it is simpler. Big national team may bring a chef and food to major events but smaller teams like the Estonian one can’t do that”, she adds. While discussing with Italian technical directors, it became evident that logistics wield a significant influence, and it is notably intricate, if not unfeasible, to predefine a precise diet for athletes in hotels or restaurants. This is particularly true within a national team such as the Italian one, where more than 100 athletes move in international competitions, each upholding individual routines encompassing both training and nutrition. Coupled with the challenges of prearranging accommodations and dining establishments, alongside potential last-minute changes, the prospect of tailoring nutritious meals to suit every athlete appears extremely complex. Bringing in a sports nutritionist for each national team could serve as a viable solution and yield a highly favorable return on investment. The nutrition expert would not only have the capacity to customize nutrition plans for individual elite athletes, in collaboration with their coaches, but also assume responsibility for managing the team’s logistical needs during transfers. Table 2 offers recommendations for addressing logistical obstacles.

## 7. Injuries Prevention and Recovery 

While fencers have a lower risk of injuries compared to football or basketball players, knee injuries are not uncommon. Furthermore, due to the frequent performance of directional shifts and lunges, there is a significant risk of muscle damage during a tournament. This risk is particularly heightened by the numerous eccentric contractions that occur during the leading leg’s foot strike [1]. Additionally, because fencers typically emphasize and develop their front muscles more than their back muscles and tend to favor one side of their body over the other, they may become susceptible to muscle strains in the less conditioned muscles. It is also important to mention that the distribution of body fat has been associated with increased injury risks involving the back, ankles, knees, joints, and muscles [21]. 

Nutrition holds notable implications for preventing injuries and expediting the recovery process, given its impact on both the athlete’s overall physical and psychological well-being, as well as its role in promoting tissue healing [52,53,54]. After prolonged and strenuous physical activity, the recovery process involves replenishing depleted energy stores, repairing damaged tissues, and starting training adaptations. Crucial factors influencing these processes include the nature, quantity, and timing of nutrient consumption, as highlighted above. Muscle damage is not solely confined to the exercise period; it can persist for several hours post-exercise. This is attributed to an extended exercise-related hormonal environment, heightened free radical levels, and acute inflammation. This tissue damage not only results in delayed onset of muscle soreness, which adversely affects performance, but also impacts the restoration of muscle glycogen and limits muscle training adaptations. The competitive demands of sports such as fencing often necessitate athletes to engage in multiple daily training sessions. Furthermore, many athletes must participate in various competitions over consecutive days, and, in some cases, even within the same day. During intense exercise, active muscles typically experience damage, which can extend beyond the workout due to an acceleration in protein breakdown. Achieving complete recovery demands initiating protein synthesis while constraining protein degradation, all affected by the nature, quantity, and timing of nutrient consumption. After exercise, elevating plasma insulin levels becomes pivotal in curbing prolonged muscle damage and triggering protein buildup. This is because insulin heightens the uptake of muscle amino acids and facilitates protein synthesis, while concurrently reducing protein degradation. For substantial post-exercise muscle protein synthesis, a well-rounded meal comprising protein and high-glycemic carbohydrates proves effective, whereas a meal exclusively composed of carbohydrates falls short. Early ingestion of a carbohydrate/protein supplement after exercise augments protein accumulation (alongside muscle glycogen replenishment), offering an extra benefit by curbing post-exercise muscle damage and fostering muscle protein buildup. These processes wield a considerable influence on subsequent exercise performance [29,55]. 

Beyond addressing minor post-training injuries, effective nutritional strategies can play a role in expediting the recovery from more serious injuries. Several nutritional tactics applicable to fencing involve increasing the daily intake of protein (mainly leucine), creatine, and omega-3 supplements to diminish muscle loss. In scenarios involving joint, ligament, and tendon injuries, strategies like consuming collagen supplements or supplements that promote its synthesis, such as vitamin C, are important for supporting healing [52,53,54,56]. Research also suggests the potential of vitamin D, calcium, and protein supplementation to assist in healing bone injuries. Additionally, in pre-operative conditions, characterized by high levels of inflammation and surgical stress inducing catabolic reactions, nutritional strategies can be beneficial. While nutrition interventions are often ignored in rehabilitation protocols, providing nutrition guidance during this process plays a crucial role in enhancing recovery [52,53,54]. Table 3 offers an outline of nutrition strategies facilitating the recovery process following injuries. It is important to note that further research is required to establish optimal protocols and validate the findings.

## 8. Limitations

The insights gathered, aside from the literature research, are derived from coaches and elite athletes within the European region, with a prominent focus on the Italian fencing team. It is worth noting that barriers to implementing nutritional counseling and the prevailing nutritional approach adopted by the national fencing team may diverge when examining other countries.

## 9. Conclusions

Tailored nutritional guidance holds paramount significance for elite fencers, aiming to elevate performance levels and expedite recovery and healing processes post-injury. Over the years, considerable efforts have been undertaken by some national fencing federations to enhance nutritional awareness. While sporadic general nutritional counseling is offered by the federations, personalized nutritional programs are not provided to athletes unless they independently seek out sports nutritionists. Furthermore, even when a personalized nutritional program is provided, logistical challenges significantly hinder its effective implementation during competitions and camps. This review offers, for the first time, a comprehensive and easy-to-use tool to assist coaches and athletes not only during and after training but also throughout competitions, as well as during injury recovery. It also provides practical recommendations to overcome logistical constraints and optimize performance in different settings.

## Figures and Tables

**Table 1 nutrients-16-01104-t001:** Nutritional guidelines for elite fencers. The guidelines draw from scientific studies and the author’s extensive experience collaborating with elite fencers over the years.

Quantity	Food Examples	Meals’ Timing	Suggestions—During Competition	Suggestions—Post-Exercise	Additional Suggestions
** *Carbohydrates* **					
7–11 g/kg of BW daily; privilege min 11 g/kg/BW daily during loading phase.	Whole grains, fruits, and vegetables; privilege rye bread for sustained energy and maintain appropriate body weight.	3–4 meals a day (carbs together with protein/fat); to be consumed within short time (max 60 min) after exercise.	Consume multiple small meals containing easily digestible carbohydrates, proteins, and fats to fuel the body and to prevent gastrointestinal discomfort (dry meat/smoked salmon/parmesan cheese/eggs plus whole grain bread plus bananas, apples/nuts/dried fruits) between bouts, as required and feasible; do not try unfamiliar foods during competitions.	Recommended carbs intake 5–7 g/kg/day for moderate-volume training; 7–10 g/kg/day for high-volume training. Consume carbs together with proteins (ratio 4:1) after exercise to restore glycogen stores and facilitate recovery from injuries, within 60 min after exercise.	Be aware of the amount and type of carbs if you need to control your weight.
** *Proteins* **					
Normally 1.5–2 g/kg of BW daily; 1.8–2.7 g/kg of BW daily if following a weight-loss diet.	Lean protein sources (poultry, fish, eggs, dairy products, tofu, and legumes); red meat 3 times a week.	3–4 meals a day (proteins together with carbs/fat); to be consumed within short time (max 60 min) after exercise.	See above	Consume proteins together with carbs (see above).	
** *Fats* **					
25% to 30% of the total energy intake—main focus on unsaturated fat (Omega 3).	Nuts, seeds, vegetable oils (flaxseed oil, walnut oil, olive oil, etc.), sea fish/oil, avocado.	3–4 meals a day (fats together with protein/carbs); to be consumed within short time (max 60 min) after exercise.	See above		Be aware of the amount of fats if you need to control your weight.
** *Fluids* **					
Around 0.5–2 L per hour during exercise; min 35 mL/kg of BW daily at rest.	Water or glucose–electrolyte solutions.	Drink throughout the day; adapt the quantity to the context (temperature, exercise intensity, etc.); do not wait to be thirsty to drink.	Drink throughout the day; privilege glucose–electrolyte solutions; make sure to consume small quantities each time to prevent gastrointestinal discomfort.	Drink as much as needed to replace fluid losses; be aware of your sweat rate.	Prioritize water consumption; consider glucose solutions, if necessary, but be mindful of their content—to be checked with a sport nutritionist.
** *Supplements* **					
To be agreed with the sport nutritionist based on individual needs and cost–benefit analyses—potential supplements: caffeine, vitamin C, iron, etc.					

**Table 2 nutrients-16-01104-t002:** Outline of practical recommendations for addressing logistical obstacles during competitions and camps. These recommendations are to be applied in the absence of healthy food or athlete-specific meals in case they cannot be pre-arranged with the hosting facilities.

Possible Strategies
Bring your own food, e.g., whole-grain rye crackers and dried fruit as source of carbohydrates, and parmesan cheese and dried meat as proteins (easy to handle and to store during long trips). Buy/pick on-site fruit such as apples, bananas, avocados, and ready-to-eat vegetables that you are used to eating. For long-term competitions such as the Olympics, you may consider bringing a steamer with you or renting a flat with a steamer and cooking your own food. To make cooking food easier, it is preferable to stay in a flat with a kitchen, rather than in a hotel. To support the above suggestions, you could consider sharing the flat and sharing shopping and meal preparation with other athletes. If the above suggestions are not possible and you have to eat at the hotel’s buffet, choose/ask for simple foods that are usually available, such as rice (brown rice at best), boiled eggs, omelets, steamed/grilled poultry, milk, whole-grain cereals, fruits, and vegetables. Avoid sauces, sweets, and what you do not recognize, especially if you are in a foreign country, to avoid gastrointestinal upset. Depending on the country, make sure you drink water from a safe source.

**Table 3 nutrients-16-01104-t003:** Outline of nutrition strategies facilitating the recovery process following injuries. The table describes four distinct scenarios: injuries resulting in muscle loss, encompassing the catabolic state before and after surgery; injuries affecting connective tissues; bone injuries; and micro-injuries between exercises.

Type of Injury	Possible Strategies
**Injuries leading to muscle loss and pre–post-surgery**	Increasing daily protein intake (mainly leucine—2.3 g/kg/day) [54] Supplementing creatine monohydrate (from 20 g down to 5 g daily) [56] Supplementing omega-3 (3 g omega-3 for 4 weeks may decrease muscle damage) [53] 24 weeks of supplementation with gelatin (10 g/day) plus vitamin C (250 mg/day) plus creatine (loading: 20 g × 5 days immediately postoperative, then 3 g/day) plus leucine (3 g/day) may attenuate leg strength loss after surgery [54]
**Joint, tendon, and ligament injuries**	Consuming 25 mL of liquid collagen containing 10 g of hydrolyzed collagen per day for 24 weeks [54] Consuming 15 g of gelatin or hydrolyzed collagen 30–60 min before exercise [54] Consuming supplements promoting collagen synthesis, such as vitamin C. Minimum intake ≥46 mg/day; a daily intake of 500 to 2000 mg of vitamin C is advised to support overall health and minimize oxidative stress [44,52,57,58]
**Bone injuries**	Consuming 800 IU/day of vitamin D and 2000 mg of calcium can reduce the risk of developing stress fractures [44,51,54] Increasing daily protein intake [54]
**Reduce micro-injuries between exercises**	Consuming short-term (4 days) high-dose vitamin C and E supplementation (vitamin C, 2000 mg/d; vitamin E, 1400 U/d) [54]

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
