# Peer review of "Nutrition for European Elite Fencers: A Practical Tool for Coaches and Athletes"

_nutrients, 2024, doi:10.3390/nu16081104_

Round 1

Reviewer 1 Report

Comments and Suggestions for Authors

The author proposes an interesting and little-considered topic in the scientific literature.

- Although interesting, I think the athletes' quotes are too frequent

- In the cognitive function I would remember creatine (10.2174/0118761429272915231122112748) and beta carotene (10.3390/brainsci13101468)

- Since the studies are few and probably need to be optimized, an opinion with practical indications from the author would be useful, perhaps justified by a scientific rationale

- In the recommendations, I would keep in mind the glycemic impact that large quantities of carbohydrates can have, even if they are wholemeal

- The concern regarding fish seems irrelevant to me, the contaminants are now present in far too many foods!

-Especially in Europe, I would consider olive oil as a source of fat

- The paragraph on supplements is too sparse, more and better should be considered, for example Vitamin B12 could be very useful, given the importance of response times

- In the table, the indications are too general, "increase protein consumption"? By how much? Vitamin C over 50mg, but at least 10 times as much! How many Omega3s, which ones?

The topic is interesting but needs to be explored in depth, so it becomes too colloquial

Comments on the Quality of English Language

It needs revision

Author Response

Thank you for dedicating your time to review this manuscript. Below are detailed responses alongside the respective revisions and corrections that have been highlighted in red within the re-submitted document:

  1. An English native speaker has reviewed the paper to enhance the language quality.
  2. A concise paragraph detailing beta-carotene and creatine has been added under "Supplements".
  3. A clearer explanation regarding the glycemic impact of consuming large quantities of carbohydrates has been added to the "Nutrients" section.
  4. Information relating to food contaminants has been removed as suggested.
  5. Olive oil has been introduced as an illustrative example.
  6. The section on supplements has been expanded with additional entries, references and precise recommendations.
  7. Table 3 has received updates to include more comprehensive details when available in the literature. Please consider that very few data are available for the elite athlete population and more investigation is needed.
  8. The author appreciates the suggestions about the citations of athletes; the author believes that these insights are crucial, offering valuable practical information for formulating nutritional strategies for elite fencers. This is particularly significant given the limited literature available on nutrition for this specific group.

Once again, I greatly appreciate your invaluable suggestions and the opportunity to enhance the manuscript accordingly.

Reviewer 2 Report

Comments and Suggestions for Authors

A review of the manuscript submitted to Nutrients entitled “Nutrition for European Elite
Fencers: A Practical Tool for Coaches and Athletes”
The paper is quite interesting and novel, yet it is not a research paper or a review but
a practical guideline based on the latest scientific data. The paper is written in good English,
and it will require only a minor spell check. The objective is clearly stated, and only the key
words must be changed, as most of them appear in the title (fencing, elite fencer, nutrition).
The introduction is much too short and does not include the metabolic demands of fencing.
The authors have placed some of these details later in the text on pages 5 and 6, while the
reviewer feels it should be in the first part to give a precise motor and energetic characteristic
of the sport discipline. It was very wise of the authors to discriminate between training and
competition, as fencing presents completely different metabolic demands during high-level
competition. Energy demands of 8–10 hour tournaments. Sweating rate and fluid loss, types
of motor actions, and muscular contractions should be presented at first, as they significantly
affect the nutritional approach. Changes in acid-base balance with post-exercise and post-
competition lactate concentrations may be included. Perhaps indicators of muscle damage
(CK, LDH) could be added to signify the enormous competition load. The training and
competition load loading is described quite well, yet the typical attacking phase should be
stressed as the foil fencers actions last 3 to 5 s, during which the ATP demand is covered
through the phosphagen system, while the epee fencers actions last much longer (up to 15 s)
and include a significant contribution of anaerobic glycolysis. This specific metabolic aspect
should be considered.
The stress and fatigue part OF THE PAPER IS PERHAPS THE MOST SIGNIFICANT ONE,
DURING WHICH THE AUTHORS STRESS THE MENTAL DEMANDS OF COMPETITION WHICH
OFTEN LAST FOR 8-10H. Nutrition and especially hydration play a pivotal role during
competition to enhance recovery between bouts. The most important aspect of nutrition
during fencing competition should be directed toward periodizing energy and proper fluids to
enhance mental recovery and maintain concentration in subsequent bouts. Considering the
specificity of fencing, both in regards to training and competition, Part 5 of the manuscript is
highly novel and very significant for sports practice. In the first part, the authors stress the role
of nutrition in fencing and its impact on cognitive function, recovery, and performance. The
authors highlight the significance nutrition has for body mass and body composition. The
nutrition section provides basic information on the macro- and micronutrient demands of
fencing, indicating the metabolic contribution of particular energy systems during
competition. Considering the great energy demands of competition and the significance of
bout recovery, the authors indicate the need for aerobic capacity, which enhances the
utilization of lactate and other post-exercise metabolites. The most novel aspect of the paper
includes the timing of meals and supplements, which greatly affect recovery and performance.
Hydration may be one of the most important aspects of nutrition affecting sport performance
in fencing, considering that protective gear limits evaporation, and dehydration during all-day
competition could exceed 2% BM. Perhaps a line or two could be added regarding the need
to monitor body hydration before and after competition, using, for example, the BHI. I also
suggest adding more details on specific guidelines for fluid intake, including its volume, timing,
and composition. While discussing supplements, I suggest including those that enhance
concentration and mental recovery, with doses and timing in regards to training, competition,
meal time, and sleep.
The paper is especially valuable because most of the data has been collected from elite
international fencers participating in international competitions. I suggest minor corrections
to the text and acceptance for print. Consider citing the following papers:
1. Judge, L. W., Bellar, D. M., Popp, J. K., Craig, B. W., Schoeff, M. A., Hoover, D. L. … Al-
Nawaiseh, A. M. (2021). Hydration to Maximize Performance And Recovery: Knowledge,
Attitudes, and Behaviors Among Collegiate Track and Field Throwers. Journal of Human
Kinetics, 79, 111-122. https://doi.org/10.2478/hukin-2021-0065
2. Filip-Stachnik, A., Kaszuba, M., Dorozynski, B., Komarek, Z., Gawel, D., Del Coso, J. …
Krzysztofik, M. (2022). Acute Effects of Caffeinated Chewing Gum on Volleyball Performance
in High-Performance Female Players. Journal of Human Kinetics, 84, 92-102.
https://doi.org/10.2478/hukin-2022-0092
3. Simoncini, L., Lago-Rodríguez, Á., López-Samanes, Á., Pérez-López, A., Domínguez, R.
(2021). Effects of Nutritional Supplements on Judo‐Related Performance: A Review. Journal of
Human Kinetics, 77, 81-96. https://doi.org/10.2478/hukin-2021-0013

Comments on the Quality of English Language

The English is fine

Author Response

Thank you for dedicating your time to review this manuscript. Below are detailed responses alongside the respective revisions and corrections that have been highlighted in red within the re-submitted document:

  1. An English native speaker has reviewed the paper to enhance the language quality.
  2. Keywords have been updated to prevent overlap with the title.
  3. The introduction section has been partially expanded to offer insights into metabolic demand, sweat rate, and more. Some content fitting for the introduction has been retained or, as requested, elaborated upon in various sections. This approach facilitated the integration of data with the perspectives of coaches and athletes, minimizing repetition.
  4. A new paragraph discussing muscle damage indicators (CK, LDH) has been incorporated into the "Nutrients" section.
  5. The attacking phase is now detailed in a specific paragraph within the "Training & Competition Schedules" section.
  6. The "Hydration" section has been supplemented with further details on specific fluid intake guidelines, including volume, timing, and composition. Moreover, a paragraph on monitoring body hydration before and after competitions has been included in the same section.
  7. The "Supplements" section has been expanded with additional information and references.

Once again, I greatly appreciate your invaluable suggestions and the opportunity to enhance the manuscript accordingly.

Round 2

Reviewer 1 Report

Comments and Suggestions for Authors

Author improved the manuscript following my suggestions so i think is suitable for publication and it will be a nice contribution in the field.